# Zileuton Attenuates Acute Kidney Injury in Glycerol-Induced Rhabdomyolysis by Regulating Myeloid-Derived Suppressor Cells in Mice

**DOI:** 10.3390/ijms26178353

**Published:** 2025-08-28

**Authors:** Tae Won Lee, Eunjin Bae, Jin Hyun Kim, Myeong Hee Jung, Dong Jun Park

**Affiliations:** 1Department of Internal Medicine, Gyeongsang National University Changwon Hospital, Changwon 51472, Republic of Korea; milkey@hanmail.net (T.W.L.); delight7607@naver.com (E.B.); 2Department of Internal Medicine, Gyeongsang National University College of Medicine, Jinju 52727, Republic of Korea; 3Institute of Medical Science, Gyeongsang National University, Jinju 52828, Republic of Korea; ajini7044@hanmail.net (J.H.K.); yallang7@hanmail.net (M.H.J.); 4Biomedical Research Institute, Gyeongsang National University Hospital, Jinju 52727, Republic of Korea

**Keywords:** Zileuton, rhabdomyolysis, acute kidney injury, myeloid-derived suppressor cells, mice

## Abstract

Rhabdomyolysis is characterized by the breakdown of skeletal muscle tissue, frequently leading to acute kidney injury (AKI). Traditional conservative treatments have shown limited effectiveness in modifying the disease course, thereby necessitating targeted pharmacological approaches. Zileuton (Z), a selective inhibitor of 5-lipoxygenase (5-LOX), has demonstrated efficacy in enhancing renal function recovery in animal models of AKI induced by agents such as cisplatin, aminoglycosides, and polymyxins. The present study aimed to evaluate the therapeutic potential of a single dose of Z in mitigating rhabdomyolysis-induced AKI (RI-AKI) via modulation of myeloid-derived suppressor cells (MDSCs). Male C57BL/6 mice were assigned to four experimental groups: Sham (intraperitoneal administration of 0.9% saline), Z (single intraperitoneal injection of Z at 30 mg/kg body weight), glycerol (Gly; single intramuscular dose of 50% glycerol at 8 mL/kg), and glycerol plus Z (Z + Gly; concurrent administration of glycerol intramuscularly and Z intraperitoneally). Animals were sacrificed 24 h post-glycerol injection for analysis. Zileuton administration significantly improved renal function, as indicated by reductions in blood urea nitrogen (BUN) levels (129.7 ± 17.9 mg/dL in the Gly group versus 101.7 ± 6.8 mg/dL in the Z + Gly group, *p* < 0.05) and serum creatinine (Cr) levels (2.2 ± 0.3 mg/dL in the Gly group versus 0.9 ± 0.3 mg/dL in the Gly + Z group *p* < 0.05). Histopathological assessment revealed a marked decrease in tubular injury scores in the Z + Gly group compared to the Gly group. Molecular analyses demonstrated that Z treatment downregulated mRNA expression of macrophage-inducible C-type lectin (mincle) and associated macrophage infiltration-related factors, including Areg-1, Cx3cl1, and Cx3CR1, which were elevated 24 h following glycerol administration. Furthermore, the expression of NLRP-3, significantly upregulated post-glycerol injection, was attenuated by concurrent Z treatment. Markers of mitochondrial biogenesis, such as mitochondrial DNA (mtDNA), transcription factor A mitochondrial (TFAM), and carnitine palmitoyltransferase 1 alpha (CPT1α), were diminished 24 h after glycerol injection; however, their expression was restored upon simultaneous Z administration. Additionally, Z reduced protein levels of BNIP3, a marker of mitochondrial autophagy, while enhancing the expression of peroxisome proliferator-activated receptor gamma coactivator 1-alpha (PGC-1α), suggesting that Z ameliorates RI-AKI severity through the regulation of mitochondrial quality control mechanisms. Zileuton also decreased infiltration of CD11b(+) Gr-1(+) MDSCs and downregulated mRNA levels of MDSC-associated markers, including transforming growth factor-beta (TGF-β), arginase-1 (Arg-1), inducible nitric oxide synthase (iNOS), and iron regulatory protein 4 (Irp4), in glycerol-injured kidneys relative to controls. These markers were elevated 24 h post-glycerol injection but were normalized following concurrent Z treatment. Collectively, these findings suggest that Zileuton confers reno-protective effects in a murine model of RI-AKI, potentially through modulation of mitochondrial dynamics and suppression of MDSC-mediated inflammatory pathways. Further research is warranted to elucidate the precise mechanisms by which Z regulates MDSCs and to assess its therapeutic potential in clinical contexts.

## 1. Introduction

Rhabdomyolysis is a clinical syndrome characterized by the destruction of skeletal muscle tissue, resulting in the release of myoglobin, sarcoplasmic proteins, and electrolytes leak into the bloodstream. These substances are subsequently filtered by the glomeruli, precipitating acute kidney injury (AKI) through multiple mechanisms such as enhanced renal vasoconstriction, ischemic tubular injury, tubular obstruction, and direct cytotoxicity of the heme proteins [1,2]. Rhabdomyolysis-induced AKI (RI-AKI) exacerbates renal structural alterations such as glomerulosclerosis and fibrosis, constituting a severe and potentially fatal complication that accounts for approximately 10% of all AKI cases [2]. Epidemiological data indicate that about 10–40% of individuals with rhabdomyolysis develop varying degrees of AKI [2,3,4,5]. The mortality rate is 8% despite advanced technologies, including intravenous fluid administration and dialysis [1,6]. Consequently, there is a critical need for targeted therapeutic strategies specifically addressing rhabdomyolysis.

Emerging evidence underscores the significant involvement of both innate and adaptive immune regulators in the pathogenesis of RI-AKI [7,8,9,10,11]. For instance, macrophage infiltration has been documented in a rat model of glycerol-induced AKI [8], and mononuclear cell depletion has been associated with a reduction in lesions in RI-AKI [9]. Myeloid-derived suppressor cells (MDSCs), heterogeneous group of cells known for their capacity to suppress T-cell responses, have been identified as key modulators of immune function [12]. In mice, MDSCs are characterized by the co-expression of the myeloid lineage differentiation markers Gr1 and CD11b [12,13]. These cells comprise a unique immunoregulatory population that maintains immune homeostasis under physiological conditions. Accumulations of MDSCs have been found in other pathological conditions, such as cancers, autoimmune diseases, infection, and different types of stress. These cells may play a critical role in modulating the normal immune response. In addition to their suppressive effects on adaptive immune responses, they also regulate innate immune responses by modulating cytokine production by macrophages [13,14,15]. Despite these insights, the specific contribution of MDSCs to the pathophysiology of RI-AKI remains unexplored.

Leukotrienes (LTs) are pro-inflammatory lipid mediators that exert a potent vasoactive effect and are produced by oxidation of 5-lipoxygenase (5-LOX)-mediated arachidonic acid [16]. Zileuton (Z) is a commercially available selective 5-LOX inhibitor that blocks LT synthesis, leading to reduced inflammation, lower capillary permeability, and limited neutrophil/eosinophil aggregation [17]. Clinically, Z is approved for the management of asthma. Some beneficial effects of Z on AKI animal models, such as cisplatin, doxorubicin, and aminoglycoside, have been demonstrated previously [18,19,20]. More recently, Z has been shown to mitigate renal tubular ferroptosis in diabetic nephropathy [21] and its combination with melatonin synergistically protects against ferroptosis-mediated renal injury, primarily via activation of the AKT/mTOR/NRF2 signaling pathway [22]. However, the therapeutic potential of Z in immune-mediated renal injury, including RIAKI, has not yet been investigated. Given the central role of immune cell infiltration in the pathogenesis of RI-AKI and the immunomodulatory properties of Zileuton, we hypothesized that Z may attenuate RI-AKI by modulating MDSC activity. To the best of our knowledge, this study represents the first investigation into the effects of Z in glycerol-induced AKI model, with a particular emphasis on its interaction with MDSCs and dynamics of immune cell populations.

## 2. Results

### 2.1. Zileuton Ameliorates Gly-Induced Renal Dysfunction and Tissue Damage

Serum levels of blood urea nitrogen (BUN) and creatinine (Cr) were significantly elevated 24 h following Gly injection in the Gly-treated mice (129.7 ± 17.9 mg/dL and 2.2 ± 0.3 mg/dL, respectively). Contemporaneous Z administration abrogated the rise in serum BUN and Cr (101.7 ± 6.8 mg/dL and 0.9 ± 0.3 mg/dL, respectively; *p* < 0.05), suggesting that Z was effective in restoring renal function (Figure 1). The administration of Z alone did not alter renal function parameters.

Histological examination using hematoxylin and eosin staining revealed the characteristic changes in acute tubular necrosis in Gly-treated mice compared to the Sham group. The tubular injury score estimated by assessing tubular cell necrosis, brush border loss, flattening of proximal tubular cells, and tubular atrophy increased in Gly-treated mice. The Z treatment attenuated the tubular injury score. This score did not change in mice that received the Z-alone treatment. TUNEL staining demonstrated an increase in tubular apoptosis following Gly injection, compared to Sham group. Notably, Z treatment reduced the number of TUNEL-positive tubular epithelial cells in the kidney (Figure 2).

### 2.2. Zileuton Reduces Macrophage Infiltration

We previously reported that macrophage infiltration increases in a rhabdomyolysis rat model and that the administration of liposome-encapsulated clodronate (LEC) decreases macrophage infiltration and restores renal function and tubular damage [9]. Mincle (macrophage-inducible C-type lectin, Clec4e), which is a pattern recognition receptor, is predominantly expressed in macrophages [23]. Mincle may aggravate renal inflammation in the context of AKI by maintaining the pro-inflammatory macrophage phenotype (M1), thereby contributing to kidney injury [24,25]. To elucidate the expression profile of mincle following Gly injection in the RI-AKI mouse model, we performed mRNA analysis. The mRNA expression levels of mincle and mincle-involved macrophage infiltration-related factors, such as Areg, Cx3cl1, and Cx3CR1, were significantly upregulated 24 h post-Gly injection. Contemporaneous Z administration markedly attenuated this mRNA expression levels (Figure 3). There was no change in the Sham or the Z-only groups.

### 2.3. Zileuton Decreases the Activation of Inflammasomes

It is well established that damage-associated molecular patterns (DAMPs) activate the NLRP3 inflammasome, and prior research has indicated that myoglobin mediates AKI via the NLRP3 inflammasome activation (10). To investigate whether the amelioration of Gly-induced AKI by Z was associated with suppression of inflammatory pathways, we assessed the expression of inflammasome-related factors including NLRP3, caspase-1, interleukin (IL)-1β, and IL-18. A significant upregulation of NLRP3 expression was observed 24 h after intramuscular Gly injection; this increase was significantly attenuated by simultaneous Z administration. Similarly, caspase-1 expression was elevated following Gly injection, but it was abrogated by Z treatment. To further confirm the inhibitory effects of Z on inflammasome activation in Gly-injected mice, we also investigated the mRNA expression levels of IL-1β and IL-18, which are pro-inflammatory cytokines related to the NLRP3 inflammasome. Both IL-1β and IL-18 mRNA expression levels were increased in Gly-only mice and significantly decreased in mice receiving combined Gly and Z treatment (Figure 4).

### 2.4. Zileuton Decreases the Activation of the HMGB1 Pathway

High mobility group box 1 (HMGB1) exhibits diverse functions contingent upon its subcellular location. In response to infection and tissue damage, HMGB1 is released both actively and passively into the extracellular milieu, where it functions as a DAMP, regulating inflammation and immune responses through different receptors such as advanced glycation end-products (RAGE) and Toll-like receptor 4 (TLR4) [26]. An investigation was conducted to examine how Gly affects HMGB1 expression and its kidney receptor. Gly administration increased the expression of HMGB1, RAGE, and TLR4 mRNA. Contemporaneous Z administration significantly attenuated the expression of these mRNAs (Figure 5). There was no change in the Sham or Z-only group. This result suggests that the release of HMGB1 by myoglobin produces marked changes in renal inflammation, and that blocking HMGB1-RAGE and/or the HMGB1-TLR4 signaling pathway by Z attenuates renal damage.

### 2.5. Effects of Zileuton on Renal Mitochondrial Quality Control

Mitochondrial quality control, which is essential for maintaining a healthy and functional mitochondrial network, encompasses processes such as mitochondrial biogenesis, dynamics, and mitophagy. It is well established that peroxisome proliferator-activated receptor γ coactivator 1 α (PGC-1α) regulates mitochondrial biogenesis and mitochondrial energy metabolism, and also plays an important role in mitochondrial dynamics and mitophagy [27] Gly administration decreased the PGC-1α mRNA expression, whereas Z attenuated this response. To determine whether the protective effects of Zin in the RI-AKI mouse model were related to mitochondrial biogenesis, the mRNA levels of mtDNA, TFAM, and CPT1α were measured. The findings revealed that the mRNA levels of mtDNA, TFAM, and CPT1α significantly decreased 24 h after Gly injection; however, contemporaneous Z administration restored these levels. These results indicate that the restorative effects of Z may be linked to the enhancement of mitochondrial biogenesis (Figure 6).

To evaluate whether the improvement of Z in the RI-AKI mouse model is associated with mitophagy, the levels of PGC-1α and BNIP3 proteins were quantified by Westen Blotting. The expression of the mitophagy marker BNIP3 increased significantly 24 h after Gly injection; however, this increase was inhibited by Z administration. The expression of Bcl-2, a major anti-apoptotic protein, was markedly restored after concurrent Z administration, while Gly injection led to its suppression (Figure 7). These results suggest that the beneficial effects of Z may be linked to the inhibition of mitochondrial mitophagy and the restoration of anti-apoptotic capacity.

### 2.6. Zileuton Reduces MDSC Infiltration

In murine models, MDSCs are characterized by the expression of CD11b (integrin α-M) and myeloid differentiation antigen Gr-1 (12, 13). Immunohistochemical staining for CD11b and Gr-1 showed that Gly injection significantly elevated the population of CD11b and Gr-1-positive cells (Figure 8A).

To determine whether the improvement of Z in the RI-AKI mouse model was related to infiltration of MDSCs, mRNA expression levels of the MDSC-related markers, including TGF-*β*1, iNOS, Arg1, and Irp4, were quantified. The results revealed that mRNA expression of TGF-*β*1, iNOS, Arg1, and Irp4α was upregulated 24 h following Gly injection, while contemporaneous Z administration normalized these expression levels (Figure 8B). These results imply that the therapeutic effects of Z may be mediated through the immunosuppressive activity of MDSCs.

## 3. Discussion

This study demonstrated that the Gly-induced deterioration of renal function in the mouse kidney was accompanied by tubular toxicity and an increase in the number of apoptotic cells. Gly also increased macrophage infiltration and the expression of MDSCs in the renal parenchyma. Contemporaneous administration of Z abrogated renal dysfunction and its associated tubular toxicity. The beneficial effects of Z in RI-AKI may have originated from the regulation of MDSCs resulting in reduced inflammation, manifested by reduced mincle-involved macrophage infiltration, reduced activation of NLRP3, increased mitochondrial biogenesis, and reduced mitochondrial mitophagy.

Macrophages are believed to contribute to the pathogenesis of several experimental AKI models [28,29,30]. Concrete evidence reveals that innate and adaptive immune regulators are associated with the pathogenesis of RI-AKI [7,8,9,10,11]. We previously showed that the depletion of macrophages by LEC ameliorates Gly-induced renal injury in mice [9]. Gly administration induced marked macrophage infiltration into the kidney, whereas depletion by LEC significantly decreased the infiltration of these cells. Depletion ameliorated renal dysfunction and restored several pathological conditions caused by Gly. One study demonstrated that specific macrophage subtypes are associated with early AKI in a Gly rhabdomyolysis mouse model [11]. Inflammatory macrophages (M1) designated with F4/80^low^CD11b^high^ increase in the kidney on day 2 after Gly administration [11]. The transmembrane pattern recognition receptor mincle is expressed in innate immune cells, such as monocytes/macrophages [31,32]. Previous reports confirmed that mincle aggravated renal inflammation in the context of AKI by maintaining the pro-inflammatory macrophage phenotype [24,25]. Our results show that Gly administration increased mincle-involved macrophage infiltration and that Z attenuated this response. Mincle-expressed inflammatory macrophages (M1) play pivotal roles in the RI-AKI mouse model.

NLRP3 plays a role in forms of AKI, such as ischemic reperfusion injury and contrast-induced AKI, by producing inflammasomes [33,34]. AKI is ameliorated by NLRP3 inflammasome inhibitors or genetic depletion of inflammasome components [35,36,37]. Several studies have suggested that myoglobin mediates AKI by activating NLRP3 [8,38]. RI-AKI has been ameliorated in NLRP3 knockout (KO) mice [10,39]. HMGB1 is a non-histone nuclear protein that has multiple functions according to its intracellular location. HMGB1 is actively secreted and passively released outside of cells, where it functions as a DAMP to mediate inflammation and immune responses through different receptors or by direct uptake [39,40,41]. Activated inflammasomes mediate the release of HMGB1 by immune cells through different signaling pathways [42,43]. Our study showed that NLRP3-activated HMGB1 was accompanied by increased expression of RAGE and TLR4. Z administration attenuated the harmful effects of Gly and restored renal function by suppressing NLRP3 and HMGB1. These results demonstrate that inflammation caused by rhabdomyolysis-damaged muscle cells or myoglobin-derived heme on endothelial cells and tubular epithelial cells plays a pivotal role in RI-AKI.

Apoptosis-mediated tubular damage has been implicated in RI-AKI [8,9,10,39,44,45]. Our previous study revealed that renal injury caused by Gly is mediated via the mitochondrial pathway, as illustrated by a shift in the balance of the Bcl-2 family proteins toward apoptosis with an increase in Bax and Bad [44]. N-acetylcysteine attenuated apoptosis by reversing the direction of the apoptotic balance in the anti-apoptotic pathway with the attenuation of Bax and Bad and increases in Bcl-2 and Bcl-xL. Some studies have shown that the apoptotic index, the ratio of total TUNEL-positive tubular apoptotic cells to total normal tubular cells, increases in RI-AKI [9,44,46]. NLRP3 inflammasomes may also have a role in apoptosis because of their ability to activate the caspase-1 pathway [10,39,45]. Several lines of evidence suggest that NLRP3 regulates apoptotic cell death by modulating mitochondrial function in AKI [39,47,48]. This study also showed that apoptotic tubular cell death increased in response to Gly injection, demonstrated by TUNEL staining and lower expression of the Bcl-2 protein and accompanied by NLRP3 activation. Z abrogated tubular apoptotic death by modulating mitochondrial function to inhibit the activation of the NLRP3 inflammasome in this study. Our study demonstrated the important role of mitochondrial apoptosis in the mechanism of RI-AKI.

Common features of MDSCs include their myeloid origin, immature state, and remarkable ability to suppress T-cell responses, leading to downregulation of the adaptive immune response [13,14,15]. The accumulation of MDSCs can be presumed in some pathological conditions, such as cancers, infections, sepsis, and some autoimmune disorders. Some reports have shown that adoptive administration of MDSCs reduced renal injury in an AKI model [49], renal fibrosis in a unilateral ureter obstruction model [50], and diabetic kidney disease [51]. In addition, another study reported that baicalein ameliorates pristane-induced lupus nephritis by regulating MDSCs [52]. We also found that Z suppressed renal injury caused by Gly, as illustrated by the regulation of MDSCs, whereas an increase in the percentage of MDSCs occurred in the RI-AKI mouse model. Z reduced the percentage of MDSCs in Gly-treated mice. Therefore, Z downregulated the expansion of macrophage cells in the RI-AKI mouse model, which might be attributed to the lower percentage of MDSCs after Z administration. This result suggests that the administration of Z in RI-AKI might be linked to its ability to regulate changes in MDSCs. This study is the first to reveal the role of MDSCs in the RI-AKI model.

Mitochondrial biogenesis, including the synthesis of mtDNA-encoded proteins, imports of nuclear-encoded mitochondrial proteins, and replication of mtDNA, suggests the generation of new healthy mitochondria to meet the energy requirement and to replenish damaged mitochondria [53,54,55]. PGC-1α acts as a critical regulator of mitochondrial biogenesis via the transcriptional machinery to increase mitochondrial mass. PGC-1α is activated by stressors, such as nutrient deprivation, hypoxia, oxidant stress, or exercise, resulting in the activation of NRF-1 and NRF-2 expression, which promotes the transcription of many mitochondrial genes and the synthesis of the mitochondrial transcription factor (TFAM). Subsequently, TFAM mediates mtDNA replication and transcription. The serial PGC-1α-NRF1/2-TFAM pathway contributes to the formation of new mitochondria [56,57,58]. Accumulating evidence shows that PGC-1α is an essential element in the regulation of mitophagy through PINK1–Parkin-dependent or -independent pathways [59,60,61]. One study showed that downregulation of PGC-1α upregulates BNIP3, a regulator of mitochondrial dysfunction and mitophagy in chondrocytes, ultimately inducing clearance of damaged mitochondria [62]. Our experiment demonstrated that Z restored mitochondrial biogenesis and regulated mitophagy in the RI-AKI mouse model by activating the PGC-1α-NRF1/2-TFAM pathway. Our study revealed that a novel mechanism in the RIAKI mouse model is associated with mitochondrial biogenesis and mitophagy.

The beneficial effects of Z against various experimental AKI animal models have been reported [18,19,20]. Cisplatin-induced nephrotoxicity, including renal dysfunction and histopathological changes, was reversed by Z. It is assumed that inhibiting the 5-LOX pathway modulates the inflammatory cascade and tissue injury. Furthermore, Z has a potent anti-apoptotic and anti-oxidant action, which is 5-LOX-independent [18]. Z exerts a protective effect in AKI induced by a 3-day regimen of doxorubicin in rats by restoring anti-oxidants and decreasing reactive oxygen species [19]. Z also attenuates amikacin- and polymyxin B-associated kidney injury [20]. The anti-inflammatory properties of Z may abrogate renal toxicity because aminoglycoside/polymyxin induces a kidney inflammatory response. Aminoglycoside/polymyxins increase mitochondrial membrane permeability and loss of energy production [63,64]. In line with these studies, we confirmed that the anti-inflammatory and anti-apoptotic properties of Z ameliorated Gly-induced renal toxicity in mice.

The most significant limitation of our study is that we did not perform flow cytometry on peripheral mononuclear cells and kidney tissues to assess changes in MDSCs, which prevented us from clearly observing alterations in MDSC subpopulations. Additionally, we did not measure the serum levels of MDSC-related cytokines. We plan to conduct additional studies focusing on confirming complementary markers of MDSC activity in a RI-AKI mice model. Despite this limitation, our findings provide novel insight into the anti-inflammatory potential of Zileuton and its relevance in immune modulation during RI-AKI.

## 4. Materials and Methods

### 4.1. Ethics Statement

All experimental procedures involving animals were reviewed and approved by the Gyeongsang National University Institutional Animal Care and Gyeongsang National University Institutional Ethics Committee (GNU-241004-M0193). The methods were carried out in accordance with the regulations and guidelines established by this committee.

### 4.2. Animal Housing

Male C57BL/6 mice (10 weeks of age) were purchased from Koatech (Suwon, Gyeonggi-do, South Korea) and maintained (3–5 mice/cage) in a temperature (23 ± 1 °C)—and humidity (55 ± 5%)—controlled facility, with a 12 h/12 h light/dark cycle. Standard mice chow and water were provided ad libitum. Mice were euthanized using CO_2_ gas when they were immobile.

### 4.3. Animal Model and Experimental Design

The dosing regimen for Z (#1724656, Sigma-Aldrich, Burlington, MA, USA) was adopted from previously published studies [19,20]. The experimental design including group allocation, was consistent with our prior investigations [65,66]. Mice were randomly assigned to one of four groups as follows: Sham group, which received an intraperitoneal (IP) injection of 0.9% saline (*n* = 3); Zileuton (Z) group, administered a single IP dose of 30 mg/kg Zileuton (*n* = 3); glycerol (Gly) group, injected intramuscularly (IM) with 50% glycerol at 8 mL/kg into the hind limb muscle (*n* = 7); and the combined group (Z + Gly), which received simultaneous IM glycerol and IP Zileuton administration (*n* = 7). All animals were euthanized 24 h following glycerol injection, and blood and tissue samples were collected for subsequent analyses.

### 4.4. Histological Analysis

Histological evaluation was conducted on kidney tissues fixed in 4% paraformaldehyde, embedded in paraffin, and sectioned at 5 μm thickness. Sections were stained with hematoxylin and eosin (H&E) to facilitate histopathological assessment. Tubular injury was semi-quantitatively scored in a blinded fashion based on criteria including epithelial flattening, loss of brush border, and luminal dilation. For each sample, 10 randomly selected, non-overlapping fields per section were examined under high magnification (×400). Injury severity was graded on a scale from 0 to 3, corresponding to the percentage of affected tubules. Image analysis was performed using NIS-Elements BR 3.2 software (Nikon, Tokyo, Japan).

### 4.5. Renal Function Assessment

Renal function was assessed by collecting whole blood via cardiac puncture from the left ventricle at the time of euthanasia. Blood urea nitrogen (BUN; #DIUR-100, BioAssay Systems, Hayward, CA, USA) and creatinine (Cr; #DICT-500, BioAssay Systems, Hayward, CA, USA) levels were quantified using commercially available colorimetric assay kits, following the manufacturers’ instructions.

### 4.6. TUNEL Assay

Apoptotic cell death was evaluated utilizing the TUNEL assay kit (#11684817910, Roche, Indianapolis, IN, USA), following established protocols [65,66]. They were semi-quantitatively scored in a blinded fashion. For each sample, 10 randomly selected, non-overlapping fields per section were examined under high magnification (×400). The enumeration of TUNEL-positive nuclei was conducted by an observer blinded to the experimental conditions, employing NIS-Elements BR 3.2 software (Nikon, Tokyo, Japan), and the mean values were calculated.

### 4.7. Quantitative Real-Time PCR (qPCR)

Tissue samples were collected for quantitative real-time PCR analysis. Total RNA was extracted from frozen kidney tissues using TRIzol (#79306, Invitrogen, Carlsbad, CA, USA). The purified RNAs were reverse-transcribed into complement DNA (cDNAs) using the iScript cDNA synthesis kit (#170-8891, Bio-Rad Laboratories, Hercules, CA, USA). Quantitative cDNA amplification was performed using a ViiA7 Real-Time System (Applied Biosystems Inc., Foster City, CA, USA), a Power SYBR Green PCR Master Mix (#4367659, Applied Biosystems), and gene-specific primers. Glyceraldehyde 3-phosphate dehydrogenase (GAPDH) served as the internal reference gene for normalization. Relative gene expression levels in each sample were quantified using the 2^−∆ΔCt^ method as previously described [24]. Primer sequences are provided in Appendix A.

### 4.8. Western Blot Analysis

Renal tissue homogenates were prepared in RIPA buffer (#89900, Thermo Fisher Scientific, USA). Equal amounts of protein (50 µg) were resolved by SDS-PAGE and subsequently transferred onto membranes. Membranes were incubated overnight at 4 °C with primary antibodies against Bcl-2 (#sc-492, Santa Cruz Biotechnology, Dallas, TX, USA), PGC-1α (ab191838, Abcam, Cambridge, UK), and BNIP3 (sc-56167, Santa Cruz Biotechnology). The detection of primary antibodies was achieved using appropriate secondary antibodies and an enhanced chemiluminescence (ECL) detection kit (Amersham Pharmacia Biotech, Piscataway, NJ, USA). The β-actin antibody (A5441, Sigma-Aldrich) was employed as a loading control. Quantitative densitometric analysis was performed in accordance with previously established methods [65,66].

### 4.9. Immunohistochemistry

Paraffin-embedded tissue sections were deparaffinized, rehydrated, and incubated with antibodies against Gr-1 (MAB1037-500, R&D systems, Minneapolis, MN, USA) and CD11b (#ab209970, Abcam). After applying biotin-conjugated secondary antibodies by the ABC Elite Kit (PK-4000, Vector Laboratories, Burlingame, CA, USA), sections were counterstained with hematoxylin to visualize cell nuclei and subsequently examined under light microscopy. Digital images were acquired and analyzed using NIS-Elements BR 3.2 software. Semiquantitative evaluation involved counting the number and density of immunohistochemically positive cells per field at ×400 magnification.

### 4.10. Statistical Analysis

Statistical analyses were performed using GraphPad Prism version 9.0 (GraphPad Software, La Jolla, CA, USA). Normality of data distribution was assessed using the Shapiro–Wilk test due to the small sample size. As the data showed a non-normal distribution, group comparisons were performed using the non-parametric Kruskal–Wallis test. Dunn’s multiple comparisons test was performed following the Kruskal–Wallis test to evaluate all possible pairwise comparisons among the four experimental groups. All data are presented as mean ± standard error of the mean (SEM) to indicate the precision of the estimated group mean in comparisons between experimental groups. A *p*-value < 0.05 was considered statistically significant.

## 5. Conclusions

The current study demonstrated the beneficial effects of Z against Gly-induced acute nephrotoxicity in mice, suggesting various mechanisms. Our experimental studies indicate that Z is a promising candidate to reduce RI-AKI but additional work is warranted.

## Figures and Tables

**Figure 1 ijms-26-08353-f001:**
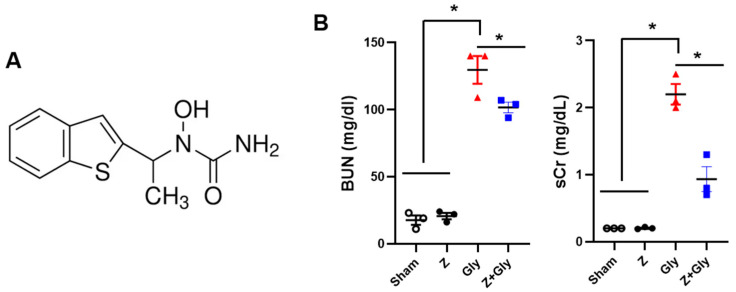
Structure of Zileuton and effects of Zileuton on renal function. (**A**) Structure of Zileuton. Mice were sacrificed 24 h after glycerol injection for blood and kidney sampling. (**B**) The blood urea nitrogen (BUN) and serum creatinine (sCr) levels were measured. Sham (*n* = 3) received 0.9% saline intraperitoneally (IP). Z (*n* = 3), received a dose of 30 mg/kg body weight. Gly (glycerol, *n* = 3), received a single dose of 50% glycerol (8 mL/kg) into a hind limb, intramuscularly. Z + Gly (glycerol plus Zileuton, *n* = 3). Statistical analysis was performed using the Shapiro–Wilk test for normality, followed by the Kruskal–Wallis test and Dunn’s post hoc test due to non-normal distribution. Data are presented as mean ± SEM to indicate the precision of the estimated group mean in comparisons between experimental groups (* *p* < 0.05). Opened circles (◦) represent the Sham group, closed circles (●) represent the Zileuton group, red triangles (▲) represent the Gly group, and blue squares (■) represent the Z + Gly group. * *p* < 0.05 vs. indicated groups.

**Figure 2 ijms-26-08353-f002:**
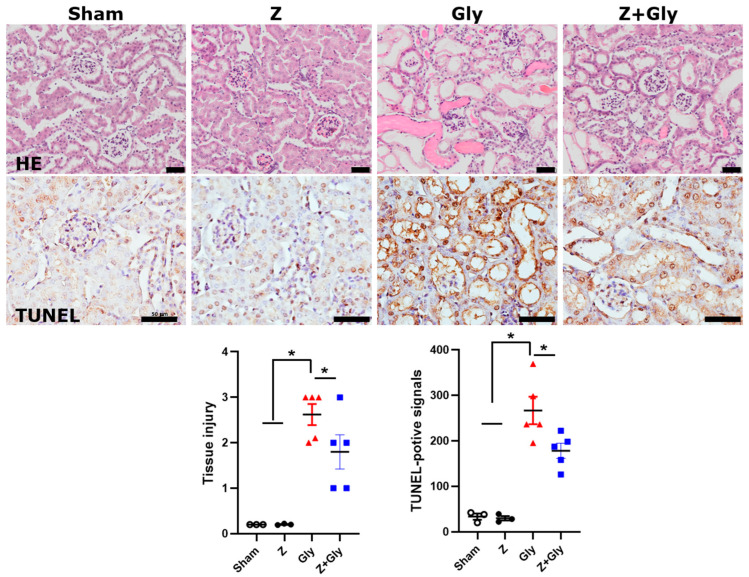
Effects of Zileuton on morphological changes and apoptosis. Histological changes were examined by hematoxylin and eosin staining and apoptotic cell death was examined by the TUNEL assay and tissue damage was quantified. Scale bar, 50 μm. Sham (*n* = 3) received 0.9% saline intraperitoneally (IP). Z (*n* = 3), received a dose of 30 mg/kg body weight. Gly (glycerol, *n* = 5), received a single dose of 50% glycerol (8 mL/kg) into a hind limb intramuscularly. Z + Gly (glycerol plus Zileuton, *n* = 5). Statistical analysis was performed using the Shapiro–Wilk test for normality, followed by the Kruskal–Wallis test and Dunn’s post hoc test due to non-normal distribution. Data are presented as mean ± SEM to indicate the precision of the estimated group mean in comparisons between experimental groups (* *p* < 0.05). Opened circles (◦) represent the Sham group, closed circles (●) represent the Zileuton group, red triangles (▲) represent the Gly group, and blue squares (■) represent the Z + Gly group. Scale bar, 50 μm. * *p* < 0.05 vs. indicated groups.

**Figure 3 ijms-26-08353-f003:**
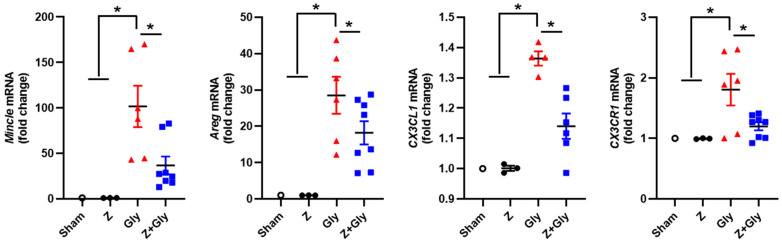
Effect of Zileuton on mincle-related macrophage infiltration in glycerol-induced kidney injury. Quantitative analyses of mincle, Areg, CX3CL1, and CX3CR1 were performed by quantitative real-time-PCR, and mRNA expression levels of each factor were normalized to the levels of GAPDH. Sham (*n* = 3) received 0.9% saline intraperitoneally (IP). Z (*n* = 3) received a dose of 30 mg/kg body weight. Gly (glycerol, *n* = 4–6) received a single dose of 50% glycerol (8 mL/kg) into a hind limb intramuscularly. Z + Gly (glycerol plus Zileuton *n* = 7). Statistical analysis was performed using the Shapiro–Wilk test for normality, followed by the Kruskal–Wallis test and Dunn’s post hoc test due to non-normal distribution. Data are presented as mean ± SEM to indicate the precision of the estimated group mean in comparisons between experimental groups (* *p* < 0.05). Opened circles (◦) represent the Sham group, closed circles (●) represent the Zileuton group, red triangles (▲) represent the Gly group, and blue squares (■) represent the Z + Gly group.* *p* < 0.05 vs. indicated groups.

**Figure 4 ijms-26-08353-f004:**
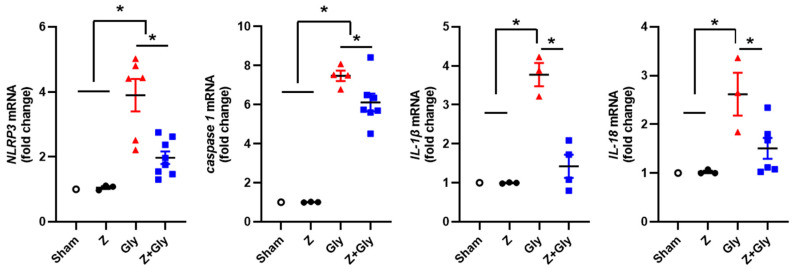
Effects of Zileuton on the expression patterns of inflammasomes. A renal extract was prepared 24 h after glycerol injection. Quantitative analyses of NLRP-3, caspase-1, IL-1b, and IL-18 were performed, and the results were normalized to the levels of GAPDH. Sham (*n* = 3) received 0.9% saline intraperitoneally (IP). Z (*n* = 3), received a dose of 30 mg/kg body weight. Gly (glycerol, *n* = 3–7), received a single dose of 50% glycerol (8 mL/kg) into a hind limb intramuscularly. Z + Gly (glycerol plus Zileuton, *n* = 4–7). Statistical analysis was performed using the Shapiro–Wilk test for normality, followed by the Kruskal–Wallis test and Dunn’s post hoc test due to non-normal distribution. Data are presented as mean ± SEM to indicate the precision of the estimated group mean in comparisons between experimental groups (* *p* < 0.05). Opened circles (◦) represent the Sham group, closed circles (●) represent the Zileuton group, red triangles (▲) represent the Gly group, and blue squares (■) represent the Z + Gly group. * *p* < 0.05 vs. indicated groups.

**Figure 5 ijms-26-08353-f005:**
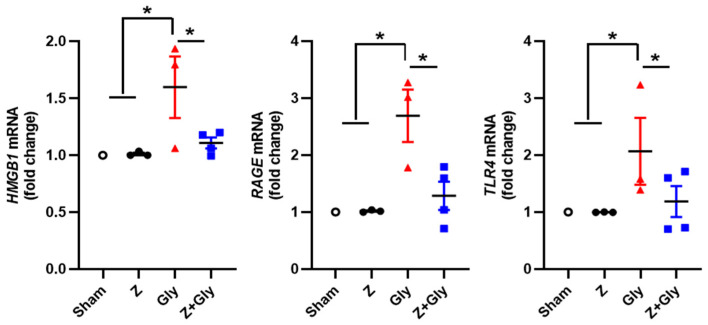
Effects of Zileuton on the HMGB1 pathway in RI-AKI mice model. A renal extract was prepared 24 h after glycerol injection. Quantitative analyses of HMGB1, RAGE, and TLR4 were performed, mRNA expression levels of each factor were measured by quantitative real-time-PCR, and the results were normalized to the levels of GAPDH. Sham (*n* = 3), received 0.9% saline intraperitoneally (IP). Z (*n* = 3), received a dose of 30 mg/kg body weight. Gly (glycerol *n* = 3), received a single dose of 50% glycerol (8 mL/kg) into a hind limb intramuscularly. Z + Gly (glycerol plus Zileuton, *n* = 4). Statistical analysis was performed using the Shapiro–Wilk test for normality, followed by the Kruskal–Wallis test and Dunn’s post hoc test due to non-normal distribution. Data are presented as mean ± SEM to indicate the precision of the estimated group mean in comparisons between experimental groups (* *p* < 0.05). Opened circles (◦) represent the Sham group, closed circles (●) represent the Zileuton group, red triangles (▲) represent the Gly group, and blue squares (■) represent the Z + Gly group. * *p* < 0.05 vs. indicated groups.

**Figure 6 ijms-26-08353-f006:**
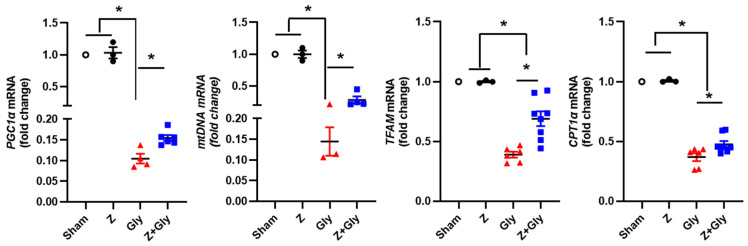
Effects of Zileuton on the mitochondrial biogenesis in RI-AKI mice model. A renal extract was prepared 24 h after glycerol injection. Quantitative analyses of PGC-1α, mtDNA, TFAM, and CPT1 α were performed, mRNA expression levels of each factor were measured by quantitative real-time-PCR, and the results were normalized to the levels of GAPDH. Sham (*n* = 3) received 0.9% saline intraperitoneally (IP). Z (*n* = 3) received a dose of 30 mg/kg body weight. Gly (glycerol, *n* = 4–7), received a single dose of 50% glycerol (8 mL/kg) into a hind limb intramuscularly. Z + Gly (glycerol plus Zileuton, *n* = 7). Statistical analysis was performed using the Shapiro–Wilk test for normality, followed by the Kruskal–Wallis test and Dunn’s post hoc test due to non-normal distribution. Data are presented as mean ± SEM to indicate the precision of the estimated group mean in comparisons between experimental groups (* *p* < 0.05). Opened circles (◦) represent the Sham group, closed circles (●) represent the Zileuton group, red triangles (▲) represent the Gly group, and blue squares (■) represent the Z + Gly group. * *p* < 0.05 vs. indicated groups.

**Figure 7 ijms-26-08353-f007:**
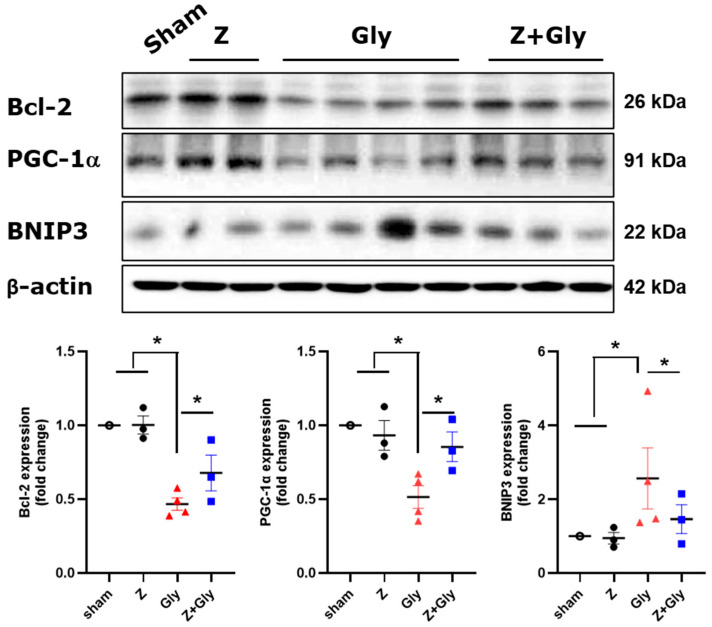
Effects of Zileuton on mitophagy in the RI-AKI mice model. A renal extract was prepared 24 h after glycerol injection. Representative expression levels of Bcl-2, PGC-1, and BNIP3 were analyzed by immunoblotting. Quantitative analyses of Bcl-2, PGC-1a, and BNIP3 were performed, and the results were normalized to the levels of β-actin Sham (*n* = 3) received 0.9% saline intraperitoneally (IP). Z (*n* = 3), received a dose of 30 mg/kg body weight. Gly (glycerol, *n* = 7) received a single dose of 50% glycerol (8 mL/kg) into a hind limb intramuscularly. Z + Gly (glycerol plus Zileuton, *n* = 7). Statistical analysis was performed using the Shapiro–Wilk test for normality, followed by the Kruskal–Wallis test and Dunn’s post hoc test due to non-normal distribution. Data are presented as mean ± SEM to indicate the precision of the estimated group mean in comparisons between experimental groups (* *p* < 0.05). Opened circles (◦) represent the Sham group, closed circles (●) represent the Zileuton group, red triangles (▲) represent the Gly group, and blue squares (■) represent the Z + Gly group. * *p* < 0.05 vs. indicated groups.

**Figure 8 ijms-26-08353-f008:**
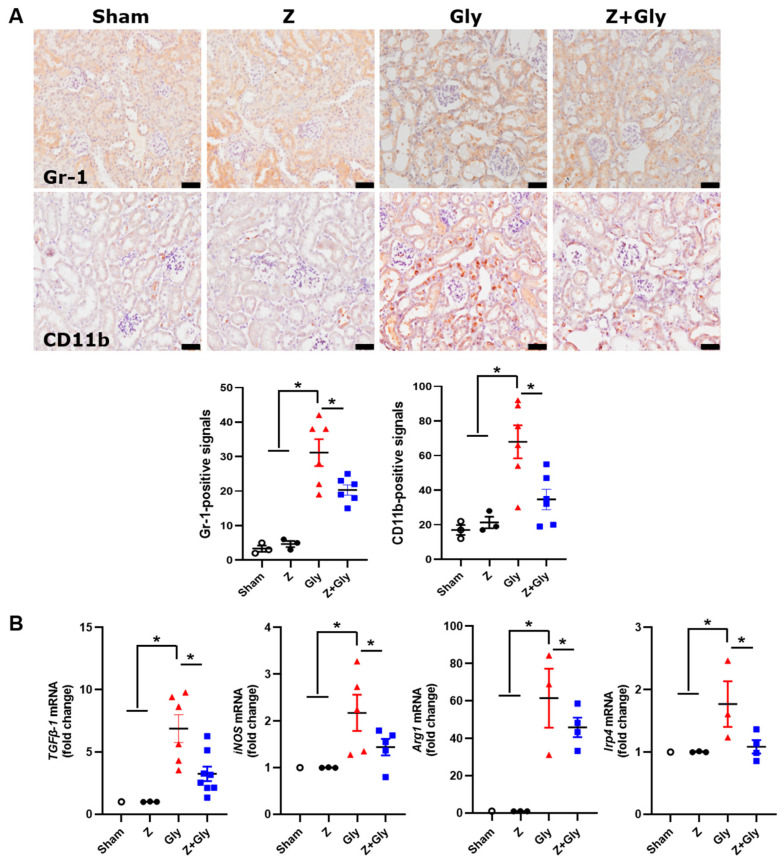
**A.** Effects of Zileuton on MDSCs infiltration in RI-AKI mice model. (**A**) Sections were stained with anti-Gr-1 and CD11b as MDSCs markers. Signals were analyzed by densitometry. Scale bar, 50 μm. Quantitative analyses of TGF-b1, iNOS, Arg1, and Irp4, MDSCs-related factors, were performed. (**B**) The mRNA expression levels of each factor were measured by quantitative real-time-PCR, and the results were normalized to the levels of GAPDH. Sham (*n* = 3) received 0.9% saline intraperitoneally (IP). Z (*n* = 3) received a dose of 30 mg/kg body weight. Gly (glycerol, *n* = 3–7) received a single dose of 50% glycerol (8 mL/kg) into a hind limb intramuscularly. Z + Gly (glycerol plus Zileuton, *n* = 4–7). Statistical analysis was performed using the Shapiro–Wilk test for normality, followed by the Kruskal–Wallis test and Dunn’s post hoc test due to non-normal distribution. Data are presented as mean ± SEM to indicate the precision of the estimated group mean in comparisons between experimental groups (* *p* < 0.05). Opened circles (◦) represent the Sham group, closed circles (●) represent the Zileuton group, red triangles (▲) represent the Gly group, and blue squares (■) represent the Z + Gly group. Scale bar, 50 μm. * *p* < 0.05 vs. indicated groups.

## Data Availability

The original contributions presented in this study are included in the article. Further inquiries can be directed to the corresponding author.

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
