# Peer review of "Zileuton Attenuates Acute Kidney Injury in Glycerol-Induced Rhabdomyolysis by Regulating Myeloid-Derived Suppressor Cells in Mice"

_ijms, 2025, doi:10.3390/ijms26178353_

Round 1
Reviewer 1 Report
Comments and Suggestions for Authors
Paper titled (Zileuton attenuates acute kidney injury in glycerol-induced rhabdomyolysis by regulating myeloid-derived suppressor cells in mice) by Authors Tae Won et al. is an experimental study described the effect of Zileuton in diminishing acute kidney injury associated with rhabdomyolysis syndrome induced by glycerol in mice and claimed the mechanism of this effect was attributed to regulating myeloid-derived suppressor cells. This study conducted on mice allows us to evaluate what happens at the histological level (H&E stainig), immunohistochemiscal level and molecular level (WB analysis of the target proteins) and above all has the merit of investigating the possible physiopathogenetic mechanisms involved in the development of acute kidney damage after administration of glycerol.
The methods used in carrying out the study are adequate. The results are documented and clarified also thanks to the attached figures, which clearly show glycerol-related damage. The discussion appears clear enough. Many studies are cited.
In this paper, plagiairism is still a big concern as some parts (even in results) appear to be plagiarized. Kindly take care of them and revise them carefully
Here, I am listing some recommendations for improving the topic. please find these comments & provide a point to point reply and highlight the changes in the file and indicate at what page & line we can follow every change.
Actually the paper needs extensive revisions due to high plagiarism and check for the use of academic English.
1- My first comment is about the high % of plagiairism (48%) and 12% from one single sources.
2- Methods is completely plagiarised and this is hard to be accepted and must be rewritten
3 - Title: Zileuton attenuates acute kidney injury in glycerol-induced rhabdomyolysis by regulating myeloid-derived suppressor cells in mice) is clear and informative
4- - Abstract must be amended by some numerical values for key findings from the study.
4 - Key words:
Zileuton; rhabdomyolysis; acute kidney injury; myeloid-derived suppressor cells 44 please add (mice)
5 - Introduction: is brief and although introduces the items of the study well, did not explore the rational or novelty of the study.
6 - AIm of the study should be clear and clarify what was the aim and how authors acheived it
7 - Methods section in general is too brief and plagiarized and lacks references MUST be extensively revised
8 - Experimental design : please give references and rationale for the selected doses of zieltuon
Please give a title for study design & describe the groups in details and regimens in a clear way
9 – Mention the number of mice in each group? how many experimental groups in the study ? and how many total animals, please indicate in a clear way
10 - What was the age and weight of the animals at the begin of the study
11 - Authors should give the source of chemicals, kits and antibodies completely and consistently (code, company, town, state and country) & version for software
12 . Statistical tests and sample sizes for each experiment should be explicitly mentioned in the methods and figure legends.
13 - Authors have to check the normality of distribution of the results by a suitable post hoc test (such as Shapiro-Wilk test or K-S test) before deciding to choose certain ANOVA. If the normality test indicated normal dist of the data, so use one-way ANOVA, if not, use non parametric ANOVA. In all cases choose a suitable post-hoc test
14 - Authors should confirm in methods that "every possible comparison between the study groups was considered" and apply this in results
15 - Mention "n" in each illustration individually
16 - Use appropriate abbreviations for minutes, seconds...etc
17 - Every abbreviation in figures should be explained in the figure legend to be self-explanatory & stands alone.
18 - In methods, Mention in details the housing conditions and how authors were keen to reduce animal suffering
Animal details and housing should be separate from the experiment design.
20 - Animal details and housing should be written in details (cage type, number per cage, food, dark light cycles, how minimized animal suffering...etc)
21 - Ensure every abbreviation is explained at the first appearance in abstract & then in the body text
22 - Methods in general lacks references
23- Results are will described and presented in a nice way
24 -Please write the limitations of this study and future directions after which
25- Page 11, line 310, please add the reference for the delta CT method for quantifying the gene expression
26- Page 11: the WB analysis method is completely plagiarized , please revise it and write the source of each chemical or antibody or kit
27- also TUNEL assay and Histopathology methods
28- In all figures, either use Zileuton or Z (be consistent please ), better than using a combination of these 2 names
29- Figures are nicely presented showing every point in the raw data
30- Authors decided to use Mean +-SEM, please explain why you decided not to use The SDM although the number of mice in each group did not exceed 30.
With the above revisions, I believe your manuscript will make a valuable contribution to the field. I encourage you to address these suggestions to improve the clarity and overall impact of your paper.
Reviewer 2 Report
Comments and Suggestions for Authors
The authors presented interesting research on the use of Zileuton, demonstrating its ability to alleviate acute kidney injury in glycerol-induced rhabdomyolysis 2 by regulating myeloid suppressor cells 3 in mice. The work is consistent with its title. The experiments are well described, and the authors draw valuable and positive conclusions.
The introduction to the project is somewhat disappointing, based on only 20 references. The authors should expand on it based on the most recent references. A figure showing the structure of Zileuton and the latest information on its activities should be included.
I suggest that the authors also prepare a supplement to the manuscript which should include the original photos currently included in the Original Images for Blots/Gels section.
Round 2
Reviewer 1 Report
Comments and Suggestions for Authors
The revised version of paper titled (Zileuton attenuates acute kidney injury in glycerol-induced rhabdomyolysis by regulating myeloid-derived suppressor cells in mice));by authors Tae Won Lee was adequately revised by the authors following the reviewer ss comments. I am glad to send a recommendation to accept this revised version
